

# The characteristics of cognitive and daily living functions of neurocognitive disorders with delusions in elderly Alzheimer's disease

Seo Yoo Kim[1,2] and Soo Jin Lee[3]

[1] Department of Psychology, Kyungpook National University, Daegu, Republic of South Korea
[2] Department of Neuropsychiatry, Good Samsun Hospital, Busan, Republic of South Korea
[3] Department of Psychology, Kyungsung University, Busan, Republic of South Korea

## ABSTRACT

**Background**. Delusions in neurocognitive disorder due to Alzheimer's disease (AD) worsen patients' cognitive functions and activities of daily living (ADL), increasing caregiver burden and the risk of mortality. AD patients with delusions tend to experience a more rapid decline in cognition and have demonstrated poorer performance on various cognitive function tests. Considering the prognosis of delusion in AD patients, it tends to be more favorable with appropriate treatment. However, there is a lack of neuropsychological research, specifically examining the impact of delusions in AD, characterized by progressive deterioration of cognitive function. This study investigates the impact of delusions on cognitive function and ADL under conditions controlling for disease severity.

**Methods**. We compared cognitive function and ADL in AD patients aged 65 years or older according to the presence of delusions. To assess longitudinal change, we analyzed data from patients monitored for an average of 15 to 16 months. We assessed cognitive function and ADL using the Seoul Neuropsychological Screening Battery-Second Edition (SNSB–II) and delusions using the Neuropsychiatric Inventory (NPI). We used IBM SPSS Statistics version 25.0 for all statistical analyses. The analysis was not adjusted for multiple comparisons. We investigated how delusions impact cognitive function and ADL, controlling for age, educational level, and disease severity.

**Results**. The delusions group exhibited poorer immediate recall of verbal memory than the non-delusions group. In the follow-up evaluation, patients who developed delusions had lower baseline cognitive function than those who did not, and their language fluency declined over time. In addition, we found the presence of delusions associated with worse functional impairment in ADL as the disease progressed.

**Conclusion**. While controlling for the severity of AD, we found no significant negative impacts of delusions on most cognitive functions. Nevertheless, it is noteworthy that the immediate recall of verbal memory and the Controlled Oral Word Association Test (COWAT)_animal sensitively detected the negative impact of delusions. Furthermore, since delusions are associated with worsening ADL, we understand that delusion treatment is important for improving the quality of life for patients and caregivers.

Corresponding author
Soo Jin Lee, leesooj@gmail.com

## INTRODUCTION

Neurocognitive disorder (NCD) is a psychiatric disorder characterized by a decline in acquired cognitive functioning due to various underlying causes. The leading cause of old age NCD is Alzheimer's disease (AD) (*Alzheimer's Association, 2019*). AD is the most common degenerative brain disorder, which tends to occur more frequently with advancing age, characterized by progressive deterioration and irreversibility. The primary symptom is impaired memory, especially in the early stages, affecting the ability to learn and retain new information. As the disease progresses, various cognitive functions such as calculation, visuospatial function, and judgment are impaired. In addition, a significant number of individuals also experience various symptoms, such as delusions, hallucinations, depression, agitation, and irritability. These non-cognitive symptoms are called behavioral and psychological symptoms of dementia (BPSD) (*O'Connell, Fenn & Hitching, 2019*; *Zhou et al., 2019*). Among BPSD, delusions are a characteristic more frequently observed in AD compared to other types of NCD (*Cummings, 2020*; *Majer et al., 2019*). In AD, delusions typically exhibit a prevalence of around 31% (27–35%) (*Zhao et al., 2016*), and as the disease progresses, the prevalence of delusions also increases (*Lai et al., 2019*).

A delusion signifies a false belief that remains steadfast despite being logically implausible or uncorrected, deviating from the cultural backdrop. AD patients with delusions exhibit exacerbated cognitive function and activities of daily living (ADL) impairments (*Connors et al., 2018*; *D'Onofrio et al., 2016*; *Majer et al., 2020*; *Poulin et al., 2017*). ADL refers to the skills individuals require to maintain their everyday activities, encompassing a comprehensive range of functions, from managing complex tasks like financial matters to basic functions such as bathing and independent mobility (*Desai, Grossberg & Sheth, 2004*). AD with delusions would amplify caregiver burden and be associated with an increased risk of mortality. Nonetheless, there is clinical significance in observing that appropriate therapeutic interventions for delusions yield more favorable responses than cognitive interventions alone (*Baharudin et al., 2019*). As AD progresses, the frequency of delusions increases. However, as cognitive impairment becomes more severe, the content of the delusions tends to become simpler and less structured. In cases of extremely severe cognitive impairment, delusions often disappear altogether. This pattern indicates that delusions are a developmental feature indicative of the progression of AD (*Ropacki & Jeste, 2005*).

Researchers have not yet fully explained the neuropathological mechanisms underlying delusions, but studies indicate that various brain regions play a role. Numerous preceding studies predominantly report a close association between delusions and the posterior cortical lesions and prefrontal functional impairments. Many previous studies suggest a strong correlation between right hemisphere lesions and frontal lobe functional impairments (*Gurin & Blum, 2017*). *Sultzer et al. (2003)* reported an association between
delusions and reduced glucose metabolism in the superior dorsolateral area and inferior frontal pole of the right frontal cortex, identified through Positron Emission Tomography (PET) analysis in AD patients. In a recent study, *Tetreault et al. (2020)* found an association between delusions in AD and atrophy in the bilateral ventrolateral frontal and superior frontal cortices. In addition, *Sultzer et al. (2014)* found AD patients with delusions exhibited reduced metabolism in the right temporal cortical regions, while *Ismail et al. (2022)* found delusions associated with atrophy in the left frontal cortex and bilateral hippocampus. Moreover, delusions in AD patients were related to functional connectivity issues with the medial temporal subcortical regions, including the hippocampus and orbitofrontal areas (*Schott et al., 2015*).

Researchers also found delusion associated with the progression of the disease (*Gottesman & Stern, 2019*). According to systematic literature reviews, when AD patients experience delusions, there is an increased risk of accelerated cognitive decline (*Modrego & Lobo, 2018*). Autopsy studies of AD patients have shown a correlation between delusions and advanced stages of neurofibrillary tangle pathology (*Ehrenberg et al., 2018*). Scholars have hypothesized that delusions may occur when there is a more pronounced underlying brain pathology at the basal level. AD patients with concomitant delusions exhibited atrophy in the frontal and medial temporal lobes in baseline examinations (*Nakaaki et al., 2012*), along with observed atrophy in various regions, including the parahippocampal gyrus, thalamus, and posterior cingulate (*Fischer et al., 2016*). *Manca et al. (2023)* also confirmed in a longitudinal study that the AD group with the onset of delusions exhibited relatively more severe atrophy in the medial temporoparietal region and subcortical and cortical areas associated with dopaminergic pathways.

In this context, the neuropathological features that demonstrate a significant association between delusions and the frontal and temporal lobes are manifested as impairments in clinical symptoms, specifically cognitive functions and ADL (*Fischer et al., 2016*; *Saari et al., 2020*). Researchers can detect neuropathological changes earlier than clinical symptoms, which manifest relatively late. However, in actual clinical settings, there is a high utility of neuropsychological assessments that offer non-invasive and objective evidence of cognitive impairment. The frontal lobe plays a crucial role in various cognitive processes, such as executive function, attention, and memory (*Chayer & Freedman, 2001*), while the medial temporal lobe is primarily associated with learning and memory functions (*Squire, Stark & Clark, 2004*).

Delusions negatively correlate with cognitive functions, including processing speed, learning ability, memory, and inferential reasoning (*Ibanez-Casas et al., 2013*). In various psychiatric disorders, cognitive functions associated with delusions are reported relatively consistently. Delusions in adults with schizophrenia spectrum disorders are linked to executive/frontal functions and memory (*Díaz-Caneja et al., 2019*). In another neurodegenerative brain disorder, Lewy body dementia, tasks reflecting executive/frontal functions were performed less effectively when delusions were present (*Tzeng et al., 2018*). Studies examining the cognitive abilities of AD patients also report a significant negative correlation between delusions and prefrontal/executive function or memory (*Fischer, Ismail & Schweizer, 2012*; *Liew, 2020*; *Na et al., 2018*). The presence of delusions is associated with

impaired performance on various cognitive function assessments, such as the Digit Span Test (DST), Trail Making Test (TMT), Rey Complex Figure Test (RCFT), and verbal memory tests (*Kumfor et al., 2022*). In a study by *Castelluccio, Malloy & McLaughlin (2020)* targeting hospitalized elderly NCD patients, the delusions group exhibited diminished performance on TMT, semantic fluency test, and verbal memory test. In addition, such cognitive impairments manifest as disturbances in ADL.

Scholars consistently highlight the importance of delusions observed in AD, driving extensive research aimed at confirming the relationship between the two to enhance our precise understanding of the disease. However, compared to neuropathological research, studies examining the correlation between delusion and cognitive functioning are relatively scarce (*Sabates et al., 2023*). In particular, there is a significant lack of longitudinal studies that have explored the impact of delusions on the cognitive characteristics of AD progression. Furthermore, previous studies mostly utilized the Mini-Mental State Examination (MMSE), a screening tool for assessing complex cognitive functions. In addition, previous research has not considered age and educational level, which are associated with cognitive impairment (*Sánchez-Izquierdo & Fernández-Ballesteros, 2021*; *Weuve et al., 2018*). Specifically, patients with delusions are more likely to have advanced disease progression (*Manca et al., 2023*), but without controlling for disease severity, functional impairment may appear more pronounced. Hence, we must actively investigate whether neuropsychological evaluation can detect the influence of delusions, even when controlling for these factors.

In this regard, our study utilizes comprehensive neuropsychological assessments and controls for age, educational level, and the severity of AD, examining whether there are differences in cognitive functions and ADL based on the presence or absence of delusions. The study identifies the cognitive functions that delusions adversely affect as the illness progresses. Furthermore, we explore whether the onset of delusions correlates with impairment in specific cognitive functions and ascertain whether there is a link between delusions and a decline in ADL.

## MATERIALS & METHODS

### Participants

This study targeted patients diagnosed with AD by specialists in the Neurology and Psychiatry departments of a hospital in Busan, South Korea, from March 2019 to September 2021 ($N = 91$). We excluded the following patients from the analysis: (1) those under 65 years of age, (2) those with a history of psychiatric treatment before receiving the AD diagnosis, (3) those unable to perform most neuropsychological assessments appropriately due to hearing or visual impairments, and (4) patients with a Clinical Dementia Rating (CDR) score of 3 or higher, considering CDR 0 for normal cognitive function and higher scores indicating more severe cognitive impairment in the disease severity assessment. After exclusions, we had 81 patients for our study, and we collected and analyzed their neuropsychological assessment results. Patients received regular prescriptions of medication tailored to their respective conditions. This study, being a retrospective study
that utilized existing data, did not require participant consent and received approval from the Kyungsung University Institutional Review Board (KSU-21-11-001).

### Procedure

Figure 1 illustrates this study's flow.

First, we divided the group of 81 patients who underwent follow-up assessments into two groups based on the presence or absence of delusions at baseline assessment. The average follow-up period for the entire group was 1 year and 3 months, during which we collected neuropsychological assessment data to categorize them into two groups based on the presence or absence of delusions. In the baseline assessment, there were 60 patients without delusions and 21 patients with delusions. Each group was examined to see if there were changes in cognitive function and ADL levels over time in follow-up assessments compared to baseline assessments. Tests showing changes over time within each group were examined to determine if there were differences between the two groups (non-delusion *vs.* delusion) based on the presence or absence of delusions.

Next, we categorized the 60 patients without delusions at the baseline assessment into two groups during the follow-up period of 1 year and 3 months: one group continued without delusions ($n = 45$), while the other group developed delusions anew ($n = 15$). Initially, we examined whether there were differences in cognitive function and ADL between the two groups (non-delusion *vs.* add-delusion) at baseline assessment. Subsequently, we investigated whether there were differences in cognitive function and ADL between baseline assessment and follow-up assessment within each group. Tests showing changes over time within each group were examined to determine if there were differences between the two groups (non-delusion *vs.* add-delusion) based on the presence or absence of delusions.

## Cognitive functions assessment

We used the Seoul Neuropsychological Screening Battery-Second Edition (SNSB–II) to assess cognitive functions. The SNSB–II is a comprehensive cognitive test battery and widely used in clinics for evaluating AD in South Korea (*Kang, Jang & Na, 2012*). The SNSB-II comprises a collection of various neuropsychological assessment tools, encompassing five cognitive domains and daily life functions, allowing for a comprehensive evaluation (*Kang, Jang & Na, 2012*). In this study, the assessments used for each cognitive domain were as follows: (1) Attention: DST, (2) Language & Related Function: Short form of the Korean-Boston Naming Test (S-K-BNT), Calculation, (3) Visuospatial Function: RCFT, (4) Memory: Seoul Verbal Learning Test (SVLT), RCFT, (5) Frontal/Executive Function: Contrasting program, Go-no go test, Controlled Oral Word Association Test (COWAT)_animal & giut ('giut' represents the Korean consonant ㄱ and corresponds to the /g/ sound in the International Phonetic Alphabet), Korean-Color Word Stroop Test (K-CWST)_60, Digit symbol coding (DSC), Korean-Trail Making Test-Elderly's version (K-TMT-E)_A & B. Numerous earlier studies have validated the reliability of the SNSB-II to detect cognitive decline for AD (*Chun et al., 2022*; *Jang et al., 2017*; *Kang et al., 2019*). For example, the reliability of the SNSB-II was reported as the test-retest method. According to the intraclass correlation coefficient analysis, the value for the subtests showed high

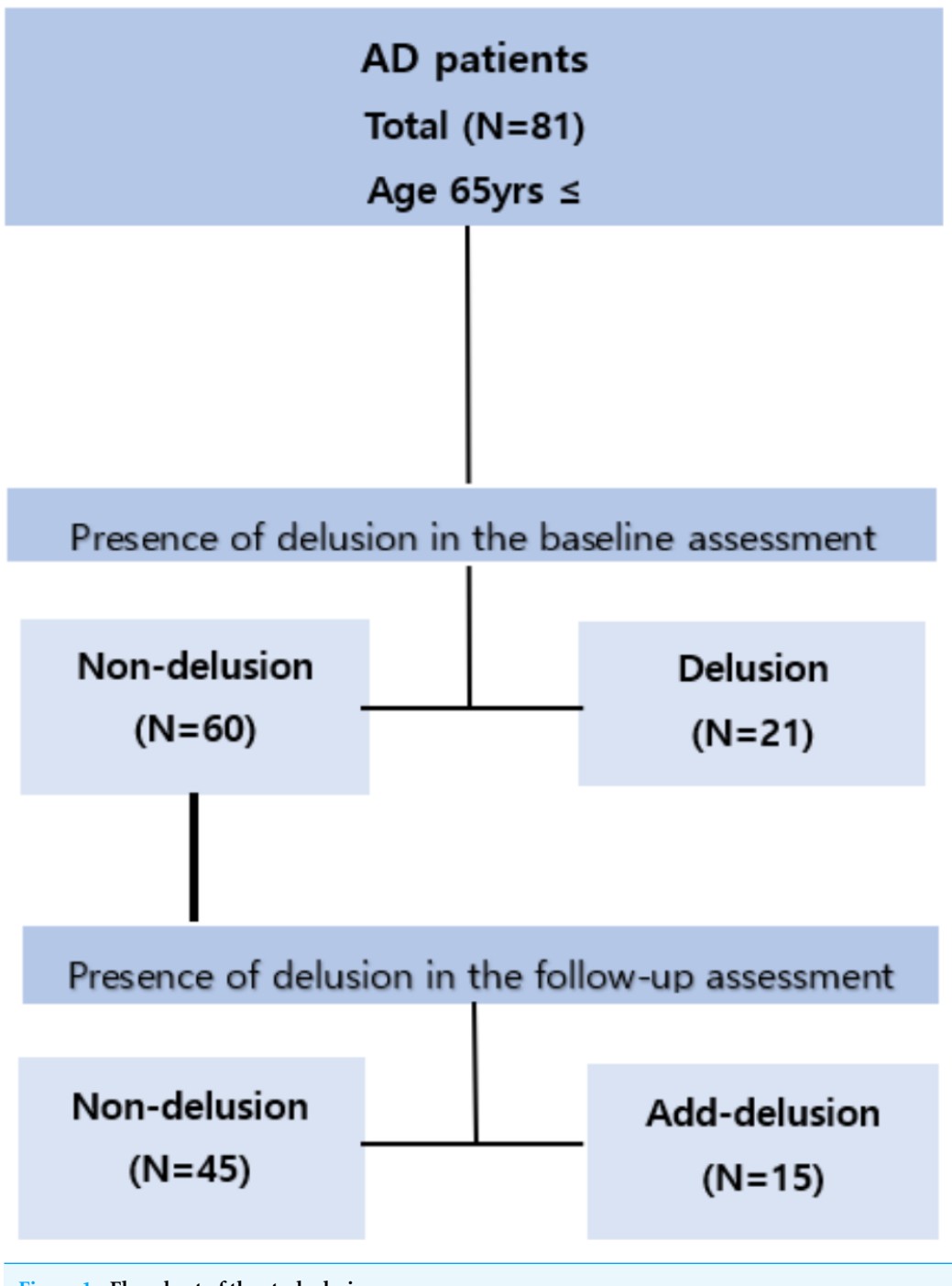

**Figure 1    Flowchart of the study design.**

significance (0.60–0.97; $p < .001$) (*Ryu & Yang, 2023*). The SNSB-II was commercially purchased from the publisher (Human Brain Research & Consulting Co., Seoul, Korea) by the hospital and we obtained the written permission from the hospital to use patients' test results for this study.

### Daily living functions assessment

We employed the Barthel Activities of Daily Living (Barthel-ADL) and the Korean-Instrumental Activities of Daily Living (K-IADL) from the SNSB-II to evaluate ADL. The Barthel-ADL consists of 10 items that assess basic ADL abilities, such as walking, dressing, and bathing independently. The optimal functional state scored 20 points, indicating a high internal consistency of 0.97 (*Kim, Won & Rho, 2004*). The K-IADL, which evaluates more advanced ADL, comprises 11 items, including tasks such as shopping, medication administration, and financial management. In contrast to the Barthel-ADL, higher scores on this scale indicate greater difficulty in independent functioning, and exhibited a high internal consistency with a coefficient of 0.96 (*Kang et al., 2002*).

### Delusions assessment

The presence of delusions was evaluated using the Neuropsychiatric Inventory (NPI). The NPI is a globally recognized instrument with established reliability and validity in various languages, making it the most widely utilized tool for assessment. The NPI is a freely available tool and commonly used in research and clinical practice. Permission for its use is not required as it is readily accessible online for academic and clinical purposes (*Cummings, 2020*). Assessment was conducted through interviews with caregivers, and it encompasses 12 domains: Delusions, Hallucinations, Agitation/Aggression, Depression/Dysphoria, Anxiety, Elation/Euphoria, Apathy/Indifference, Disinhibition, Irritability/Lability, Aberrant motor behavior, Sleep/Night-time behavior, and Appetite/Eating disorders (*Cummings et al., 1994*). Among patients with a frequency (0–4) of 1 or higher in the "Delusions" category, delusions were considered present. The Korean version of the NPI was validated for reliability and validity among patients with dementia by *Choi et al. (2000)* (Cronbach's alpha =0.85).

### Statistical procedures

To compare the demographic characteristics, average follow-up assessment period, and disease severity between two groups based on the presence or absence of delusions, we conducted an independent samples $t$-test and chi-square test. To compare cognitive function and ADL between the two groups, we conducted an ANCOVA controlling for age, education level, and disease severity. For the analysis of longitudinal data, we conducted a paired $t$-test to compare the cognitive function and ADL between the baseline and follow-up assessment within each group. The analysis was not adjusted for multiple comparisons.

We used IBM SPSS Statistics version 25.0 for all statistical analyses, and we considered a $p$-value less than 0.05 statistically significant.

## RESULTS

### Comparison according to the presence of delusion in baseline assessment

To investigate whether delusions have an impact on disease deterioration among AD patients with follow-up evaluation ($n = 81$), we divided the group based on the presence of

**Table 1  Comparisons of demographic characteristics according to the presence of delusion in the baseline visit.**

| | | Presence of delusion in the baseline assessment | | | | |
| --- | --- | --- | --- | --- | --- | --- |
| | | Non-delusion ($n = 60$, 74.1%) | Delusion ($n = 21$, 25.9%) | Total ($n = 81$, 100%) | $t$ or $\chi^2$ | $p$ |
| Age | | 77.63 ± 4.90 | 77.62 ± 5.31 | 77.63 ± 4.97 | 0.011 | 0.991 |
| Sex | female | 45 (75.0%) | 14 (66.7%) | 59 (72.8%) | 0.546 | 0.460 |
| | male | 15 (25.0%) | 7 (33.3%) | 22 (27.2%) | | |
| Education | | 3.86 ± 3.92 | 4.80 ± 3.77 | 4.14 ± 3.90 | −0.994 | 0.323 |
| CDR[*] | baseline | 0.68 ± 0.29 | 0.91 ± 0.44 | 0.74 ± 0.35 | −2.62 | 0.011 |
| | follow-up | 1.03 ± 0.85 | 1.24 ± 0.66 | 1.09 ± 0.81 | −1.00 | 0.319 |
| Duration | month | 15.4 ± 5.44 | 14.2 ± 4.37 | 15.1 ± 5.18 | 0.858 | 0.393 |

Notes.

[*]$p < 0.05$.

CDR, Clinical Dementia Rating.

delusions during the initial assessment. The delusions group ($n = 21$) and the non-delusions group ($n = 60$) did not show significant differences in sex, age, and education level, or follow-up assessment period. However, the dementia severity in the delusions group was significantly higher at the baseline assessment (Table 1). The average follow-up period for all participants was one year and three months (15.1 months).

We conducted paired t-tests to explore whether there were differences in cognitive function and ADL during the initial and follow-up assessment. The results revealed no significant differences in any assessments in the non-delusion group. Conversely, a significant difference emerged in the delusion group only in immediate verbal memory recall (SVLT immediate recall, $t = 2.581$, $df = 20$, $p = 0.010$, $d = 0.52$), despite not showing differences in most cognitive domains. Moreover, we identified significant differences between the Barthel-ADL ($t = 2.446$, $df = 20$, $p = 0.024$, $d = 0.70$) and K-IADL ($t = -5.357$, $df = 20$, $p < .000$, $d = 1.26$), both of which assess ADL.

To examine whether the three tests (SVLT, Barthel-ADL, K-IADL) that showed significant differences also demonstrated differences between the two groups, we conducted ANCOVA controlling for age, education level, and disease severity. As a result, there were no significant differences in any of the three tests based on the presence or absence of delusions at baseline assessment. In contrast, during the follow-up assessment, significant differences were observed in all three tests based on the presence or absence of delusions(SVLT immediate recall, $F = 4.008$, $p = 0.049$, $\eta^2 = 0.050$; Barthel-ADL, $F = 5.658$, $p = 0.020$, $\eta^2 = 0.069$; K-IADL, $F = 11.593$, $p = 0.001$, $\eta^2 = 0.132$) (Fig. 2).

## Comparison according to the presence of delusion in follow-up assessment

We investigated if cognitive function and ADL differed initially between patients who later developed delusions and those who did not. In the average one year and three months (15.4 months) of follow-up assessment, we divided the participants into two groups based on the presence ($n = 45$) or absence ($n = 15$) of delusions (i.e., non-delusion vs. add-delusion). We found no significant differences in demographic characteristics and disease severity between the two groups (Table 2).

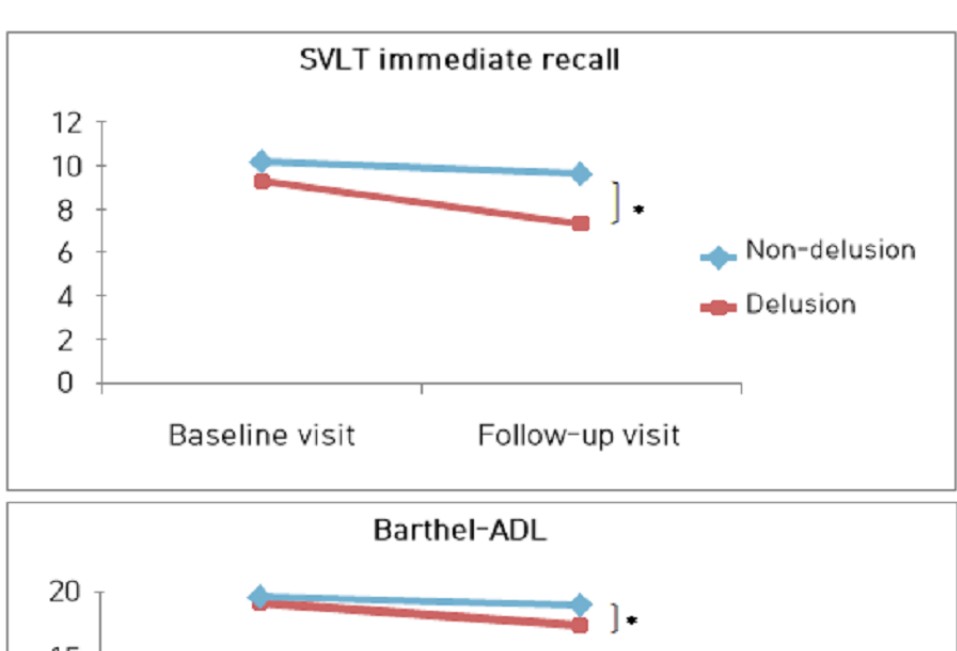

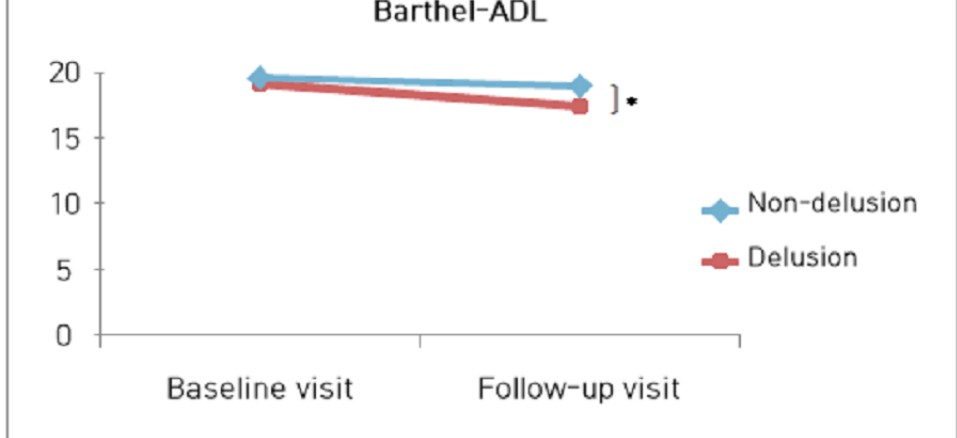

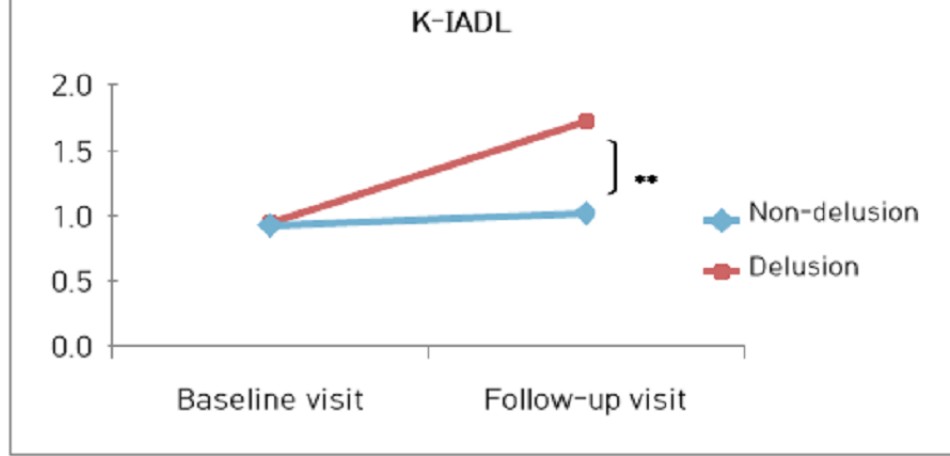

**Figure 2  Changes in cognitive and daily living functions according to the presence of delusion in the course of visits.** Over the course of time, the delusions group exhibited significant differences in cognitive functions, particularly in language and memory abilities, 

**Figure 2 (...continued)**
as evaluated by the Seoul Verbal Learning Test (SVLT)_immediate recall ($F = 4.008$, $p = 0.049$, $\eta^2 = 0.050$). Additionally, their performance in assessing ADL was notably diminished, as evidenced by the Barthel Activities of Daily Living (Barthel-ADL, $F = 5.658$, $p = 0.020$, $\eta^2 = 0.069$) and the Korean-Instrumental Activities of Daily Living (K-IADL, $F = 11.593$, $p = 0.001$, $\eta^2 = 0.132$).

**Table 2 Comparisons of demographic characteristics according to the presence of delusion in the follow-up visit.**

| | | Presence of delusion in the follow-up assessment | | | | |
|---|---|---|---|---|---|---|
| | | Non-delusion ($n = 45$, 75.0%) | Add-delusion ($n = 15$, 25.0%) | Total ($n = 60$, 100%) | $t$ or $\chi^2$ | $p$ |
| Age | | $77.53 \pm 4.93$ | $77.93 \pm 4.96$ | $77.63 \pm 4.90$ | 0.272 | 0.787 |
| Sex | female | 36 (80.0%) | 9 (60.0%) | 45 (75.0%) | 2.400 | 0.121 |
| | male | 9 (20.0%) | 6 (40.0%) | 15 (25.0%) | | |
| Education | | $3.80 \pm 3.51$ | $4.03 \pm 5.10$ | $3.86 \pm 3.92$ | $-0.198$ | 0.844 |
| CDR | baseline | $0.65 \pm 0.23$ | $0.80 \pm 0.41$ | $0.68 \pm 0.29$ | $-1.831$ | 0.072 |
| | follow-up | $0.99 \pm 0.94$ | $1.17 \pm 0.45$ | $1.03 \pm 0.85$ | $-0.700$ | 0.487 |
| Duration | month | $15.6 \pm 5.45$ | $14.6 \pm 5.51$ | $15.4 \pm 5.44$ | 0.627 | 0.533 |

**Notes.**
CDR, Clinical Dementia Rating.

To investigate whether there were differences in cognitive function and ADL during the initial evaluation based on the occurrence of delusions in the follow-up assessment, we conducted ANCOVA after controlling for age, education level, and the severity of AD. The results showed a significant difference in attention (DST Forward, $F = 9.750$, $p = 0.003$, $\eta^2 = 0.151$; DST Backward, $F = 6.939$, $p = 0.011$, $\eta^2 = 0.112$), visuospatial function (RCFT Copy, $F = 4.082$, $p = 0.048$, $\eta^2 = 0.069$), and frontal /executive function (COWAT_giut, $F = 5.681$, $p = 0.022$, $\eta^2 = 0.127$; DSC, $F = 7.391$, $p = 0.009$, $\eta^2 = 0.118$; K-TMT-E_A, $F = 6.675$, $p = 0.013$, $\eta^2 = 0.112$) (Table 3). However, the two groups had no significant differences in other cognitive domains of memory, language and related function, and ADL (Table 3).

Furthermore, we compared each group's baseline and follow-up assessments with paired $t$-tests, examining whether cognitive function and ADL changes occurred. In the findings, the non-delusions group did not exhibit significant assessment differences. In contrast, the add-delusions group displayed significant differences in the semantic fluency aspect of the frontal/executive function (COWAT_animal, $t = 2.172$, $df = 14$, $p = 0.047$, $d = 0.49$) despite not showing differences in most cognitive domains. Moreover, we identified significant differences between the Barthel-ADL ($t = 3.066$, $df = 14$, $p = 0.008$, $d = 0.94$) and K-IADL ($t = -4.521$, $df = 14$, $p < 0.000$, $d = 0.93$), both of which assess ADL.

Finally, we conducted an ANCOVA to compare groups with and without delusions in assessments showing significant differences, controlling for age, education level, and severity of AD. As a result, all three tests showed no significant differences based on the presence or absence of delusional onset at baseline assessment, but significant differences were observed during the follow-up evaluation (COWAT_animal, $F = 8.514$, $p = 0.005$,

**Table 3 Comparison of cognitive sub-test scores and ADL in baseline assessment according to the presence of delusion in follow-up visit after controlling for age, education level, and the severity of AD.**

| | | Non-delusion ($n = 45$, 75.0%) | Add-delusion ($n=15$, 25.0%) | F | p |
|---|---|---|---|---|---|
| **Attention** | **DST Forward**[**] | 4.36 ± 1.00 | 3.27 ± 1.49 | 9.750 | 0.003 |
| | **DST Backward**[*] | 2.06 ± 1.26 | 1.58 ± 1.33 | 6.939 | 0.011 |
| Language & Related Function | S-K-BNT | 8.53 ± 3.29 | 6.47 ± 3.70 | 2.797 | 0.100 |
| | Calculation | 5.36 ± 3.21 | 3.93 ± 2.94 | 2.598 | 0.113 |
| **Visuospatial Function** | **RCFT Copy**[*] | 12.72 ± 10.77 | 7.30 ± 10.24 | 4.082 | 0.048 |
| Memory | SVLT immediate recall | 10.76 ± 4.54 | 8.53 ± 4.22 | 1.056 | 0.309 |
| | SVLT delayed recall | 0.96 ± 1.85 | 0.40 ± 0.91 | 0.723 | 0.399 |
| | RCFT immediate recall | 2.48 ± 3.67 | 1.07 ± 1.72 | 1.465 | 0.231 |
| | RCFT delayed recall | 1.61 ± 2.84 | 0.50 ± 1.57 | 1.855 | 0.179 |
| **Frontal/Executive Function** | Contrasting program | 12.91 ± 8.56 | 10.47 ± 10.14 | 0.639 | 0.427 |
| | Go-no go test | 10.75 ± 8.00 | 6.73 ± 6.24 | 2.458 | 0.123 |
| | COWAT_animal | 9.62 ± 4.51 | 8.07 ± 4.33 | 0.534 | 0.468 |
| | **COWAT_giut**[*] | 3.82 ± 3.16 | 1.80 ± 2.74 | 5.681 | 0.022 |
| | K-CWST_60 | 12.50 ± 12.41 | 7.87 ± 8.75 | 1.017 | 0.318 |
| | **DSC**[**] | 16.71 ± 13.49 | 7.67 ± 11.79 | 7.391 | 0.009 |
| | **K-TMT-E_A**[*] | 55.44 ± 51.58 | 122.27 ± 113.88 | 6.675 | 0.013 |
| | K-TMT-E_B | 173.93 ± 121.28 | 213.20 ± 126.63 | 1.288 | 0.262 |
| Daily Living Function | Barthel-ADL | 19.60 ± 1.07 | 19.60 ± 1.06 | 1.609 | 0.210 |
| | K-IADL | 0.98 ± 2.93 | 0.77 ± 0.61 | 0.862 | 0.357 |

**Notes.**
[*] $p < 0.05$.
[**] $p < 0.01$.
DST, Digit Span Test; S-K-BNT, Short form of the Korean version-Boston Naming Test; RCFT, Rey Complex Figure Test; SVLT, Seoul Verbal Learning Test; COWAT, Controlled Oral Word Association; K-CWST, Korean-Color Word Stroop Test; DSC, Digit Symbol Coding; K-TMT-E, Korean-Trail Making Test-Elderly's version; Barthel-ADL, Barthel Activities of Daily Living; K-IADL, Korean-Instrumental Activities of Daily Living.

$\eta^2 = 0.134$; Barthel-ADL, $F = 7.637$, $p = 0.008$, $\eta^2 = 0.122$; K-IADL, $F = 4.497$, $p = 0.038$, $\eta^2 = 0.076$) (Fig. 3).

## DISCUSSION

This study examined the relationship between disease progression and the presence of delusions among patients with Alzheimer's disease (AD) who underwent follow-up assessments. When comparing groups based on the presence or absence of delusions, we controlled for age, education level, and severity of AD in all analyses. First, when analyzing groups based on the presence or absence of delusions during the initial assessment, the group with delusions showed a significant decline in immediate recall ability of verbal memory (*i.e.,* SVLT) and activities of daily living (ADL). AD patients commonly experience a decline in their ability to learn new information. When delusions are present, this impairment becomes more severe compared to the non-delusions group. Our brain generates predictions and learns from the errors that result from these predictions, which drives adaptive behavior by updating memories to incorporate new information (*Sinclair et al., 2021*). Such prediction errors trigger learning when there is a discrepancy between

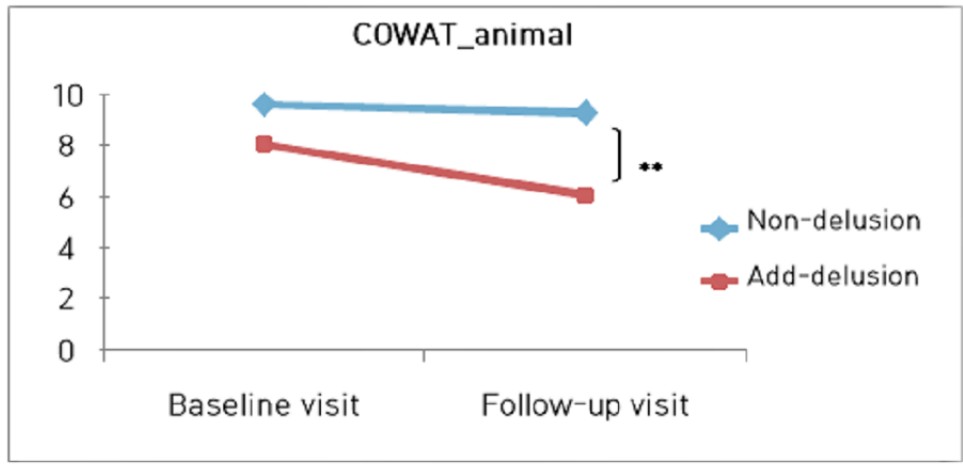

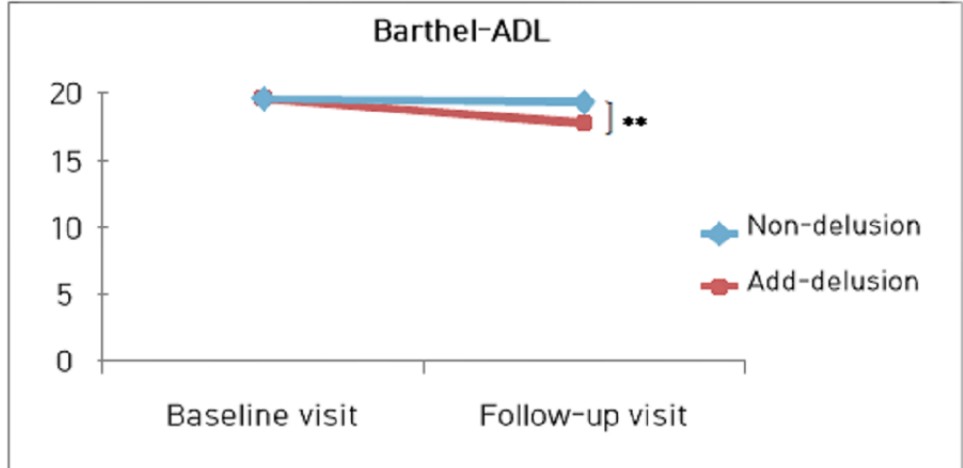

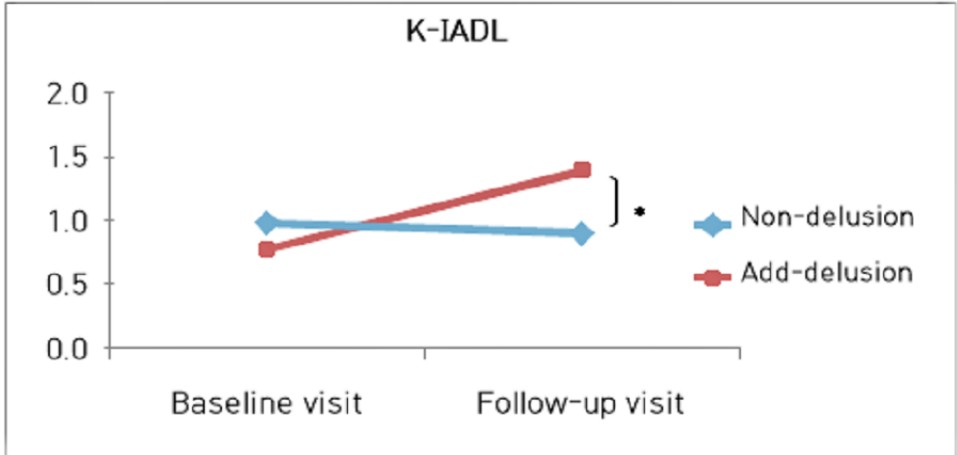

**Figure 3  Changes in cognitive and daily living functions according to the emergence of delusion in the course of visits.** In patients with Alzheimer's disease who initially did not exhibit delusions, the group that developed delusions during the follow-up visit showed a significant 

**Figure 3 (…continued)**
decline only in frontal/executive function, as assessed by the Controlled Oral Word Association Test (COWAT)_animal ($F = 8.514$, $p = 0.005$, $\eta^2 = 0.134$), compared to the group that did not experience delusions. Furthermore, their performance in evaluating ADL was notably diminished, as indicated by the Barthel Activities of Daily Living (Barthel-ADL, $F = 7.637$, $p = 0.008$, $\eta^2 = 0.122$) and the Korean-Instrumental Activities of Daily Living (K-IADL, $F = 4.497$, $p = 0.038$, $\eta^2 = 0.076$).

reality and expectations. Multiple previous studies have confirmed that delusions are associated with abnormal prediction errors (*Corlett et al., 2013*; *Fineberg & Corlett, 2016*). In other words, prediction errors are associated with the learning of information, and failures in the immediate recall task of the SVLT-E language memory test reflect a decline in the ability to learn and encode language information (*Fernaeus et al., 2014*; *Jang et al., 2019*). In essence, the findings of this study suggest that the connection between delusions and abnormal prediction errors results in decreased performance during the learning phase of different memory processes. On the other hand, the non-delusional group did not show significant differences in cognitive function over time. These findings suggest that the progression of cognitive impairment may occur very gradually, as evidenced by no significant differences observed over a average follow-up period of 1 year and 3 months. However, caution is warranted in interpreting these results due to the small sample size.

Second, we divided the groups based on whether they developed delusions during the follow-up period. We focused on individuals without initial delusions and monitored for an average of one year and three months. We examined the differences in cognitive function and ADL for both groups during the initial assessment. The results indicated that in the group where delusions were present during the follow-up assessment, there was a more pronounced impairment in attention, visuospatial function, and frontal/executive functions compared to the group without delusions. However, we observed no significant differences between the two groups regarding ADL. In essence, when delusions are present, there may not be a significant difference in the ADL before the onset of delusions. However, this state seems to be associated with relatively more severe brain damage and subsequent cognitive decline, which could potentially contribute to the development of delusions (*Manca et al., 2023*; *Nakaaki et al., 2012*). Recent research shows that cognitive function in the brain mediates cortical atrophy and delusions (*Kwak et al., 2022*). Therefore, proactive therapeutic interventions to enhance and maintain cognitive function across various domains could potentially aid in preventing the onset of delusions.

Third, we investigated whether the onset of delusions had an impact on the progression of the disease. The results indicated that when delusions emerged during the disease, there was a significant decline in semantic fluency (*i.e.,* Controlled Oral Word Association Test (COWAT)_animal) and ADL abilities compared to the group without delusions. Prior studies have shown an association between delusions and reduced language fluency (*Selva et al., 2007*; *Sun et al., 2018*), and delusions accelerate the progression of the disease and temporal lobe atrophy (*Qian et al., 2019*). Within the frontal/executive functions domain, semantic fluency is more closely associated with temporal lobe functions than phonemic fluency, particularly in language fluency. Given these considerations, it becomes clear why

the group that developed delusions as the disease progressed performed worse on the COWAT_animal task in our study.

In this study, the scenario where delusions were present along with the progression of AD and the scenario where delusions emerged during AD progression showed significant declines in ADL. *Saari et al.*'s (*2020*) study involving a five-year follow-up observation of patients with AD confirmed that delusions contribute to the deterioration of functional impairment in ADL. Functional impairment in ADL necessitates greater assistance from caregivers, increasing the burden on the caregivers. Therefore, early therapeutic intervention for delusions could help maintain the patient's daily functioning, ultimately enhancing the quality of life for patients and caregivers.

Additionally, during AD's progression, delusions deteriorated ADL impairments observable by caregivers, yet most cognitive functions did not exhibit pronounced differences in performance based on the presence of delusions. While numerous previous studies have indicated that delusions are associated with declines in various cognitive functions, our study did not demonstrate significant differences in most cognitive functions based on the presence or absence of delusions. We can attribute this phenomenon to controlling for the severity of AD in our study. AD, characterized by progressive deterioration, progresses independently of other risk factors, and the influence of these risk factors diminishes with increasing disease severity (*Steenland et al., 2012*). Numerous studies report that AD and psychiatric symptoms, including delusions, tend to exhibit independent progression patterns and often do not have significant correlations (*Eikelboom et al., 2021*; *Gottesman et al., 2021*). Delusions tend to manifest in individuals with AD after a certain stage of progression (*Liew, 2019*). Therefore, in other studies where various cognitive functions are more impaired in the presence of delusions, it is highly likely that this is due to the increased severity of AD progression. Thus, to understand the impact of delusions on cognitive impairment in AD patients, it is crucial to control for the extent of cognitive impairment caused by the disease itself. However, this aspect has not been considered in most previous studies.

Furthermore, the effect of pharmacological treatments is also considered in the observed performance differences seen in only a few tests based on the presence of delusions. In *Chang et al.*'s (*2021*) study, which involved tracking AD patients for at least one year, the absence of a significant correlation between disease progression and the presence of behavioral and psychological symptoms of dementia (BPSD) was due to the cognitive-enhancing effects of drugs such as Donepezil. Furthermore, recent research in Sweden has observed that the use of acetylcholine agonists, which are cognitive-enhancing drugs for AD patients, also has the potential to suppress various BPSD (*Tan et al., 2020*). *Matsunaga et al. (2018)* reported that Memantine, which functions as a glutamate antagonist on N-Methyl-D-aspartic acid (NMDA) receptors, demonstrated efficient improvement in cognitive function and behavioral disturbances in AD patients when used alone or in combination with Donepezil.

This study targeted patients who received regular prescriptions of cognitive enhancers tailored to their respective disease conditions. Therefore, pharmacological treatment could potentially delay cognitive decline and improve delusions without showing significant differences between the delusion and non-delusion groups. These findings underscore

the importance of conducting regular follow-up assessments of cognitive function and monitoring the presence of delusions in AD patients, highlighting the significance of implementing appropriate pharmacological interventions.

Nonetheless, the significant performance difference observed in the SVLT-E immediate recall and COWAT_animal, based on the presence or absence of delusions, is noteworthy despite the above factors. The study confirmed that SVLT-E sensitively measures deteriorating language learning abilities exacerbated by delusions. Furthermore, COWAT-animal was found to be associated with the likelihood of delusion onset. These findings suggest that both tests could be effectively utilized in detecting delusions. On the other hand, the results showing impaired performance in different cognitive domains depending on the timing of delusion presence during the disease progression may appear somewhat inconsistent. However, both tests share the commonality of assessing language function. According to previous studies, damage to the left temporal lobe has been confirmed to play a significant role in various aspects of language function (*Gleissner & Elger, 2001*). In other words, it is possible that the exacerbated functional decline due to delusions (*Qian et al., 2019*) in AD, where temporal lobe damage is prominent (*Eberling et al., 1992*), was observed in different tasks. Therefore, additional research is needed to clarify these results.

The limitations of this study are as follows: First, we did not match the demographic characteristics of the control group. Enhancing homogeneity within the control group is crucial for obtaining more precise results. Second, since the study focused on patients from a single institution in a specific region, it is challenging to consider the results completely representative of AD patients. Therefore, there are limitations to generalizability. Third, relying on caregiver reporting for assessments such as NPI and K-IADL introduces limitations. The results could vary based on the caregiver's personality traits, the nature of their relationship with the patient, and the duration of caregiving. Due to various factors influencing caregivers, they might report patient symptoms with exaggeration or underestimation. Hence, it is necessary to utilize more objective assessment tools and employ the clinician's adept interviewing skills to achieve a more objective evaluation. Lastly, AD patients commonly exhibit various BPSD assessed by the NPI, which were not accounted for in our analysis. Moreover, delusions assessed by the NPI are typically evaluated by multiplying their frequency and severity scores to yield a range from 0 to 12, but our study only dichotomized them based on presence or absence. Therefore, we recommend that future studies should conduct more refined analyses to address these methodological limitations.

## CONCLUSIONS

This study examined the differences in cognitive function and ADL based on the presence or absence of delusions in patients with AD. Significant differences across most cognitive domains were not evident when controlling for the severity of AD. However, the delusions group showed a decline in verbal learning ability over time, and the onset of delusions during AD exacerbated the decline in semantic fluency. Delusions particularly exacerbated functional impairments in activities of daily living during the

progression of AD. Additionally, the delusion-onset group was found to have more severe cognitive impairment due to AD at the onset of delusions compared to the group without delusions.The significance of this study is underscored by confirming the impact of delusions on AD patients through comprehensive neuropsychological assessments, while controlling for the severity of the disease. This study lays the association between delusions and cognitive functioning as a potential area for further investigation.

### Funding
The authors received no funding for this work.

### Competing Interests
The authors declare there are no competing interests.

### Author Contributions
- Seo Yoo Kim conceived and designed the experiments, performed the experiments, analyzed the data, prepared figures and/or tables, authored or reviewed drafts of the article, and approved the final draft.
- Soo Jin Lee conceived and designed the experiments, analyzed the data, authored or reviewed drafts of the article, and approved the final draft.

### Human Ethics
The following information was supplied relating to ethical approvals (i.e., approving body and any reference numbers):

Kyungsung University Institutional Review Board (KSU-21-11-001).

### Data Availability
The raw data are available in the Supplementary File.

### Supplemental Information
Supplemental information for this article can be found online at http://dx.doi.org/10.7717/peerj.18026#supplemental-information.

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
