# Peer review of "The characteristics of cognitive and daily living functions of neurocognitive disorders with delusions in elderly Alzheimer’s disease"

_PeerJ, doi:10.7717/peerj.18026_

## Round 0.1 · original submission · Minor Revisions

Dear Authors,

Thank you for submitting your manuscript to Peer J.

The Reviewers have highlighted a number of aspects that should be clarified. Please consider all these comments and suggestions and revise the manuscript accordingly.

Kind regards,
Marialaura Di Tella

·

Basic reporting

This study examines the neuropsychological correlates of delusions in Alzheimer's disease. The paper is well-structured and easy to follow. The literature cited in the background is appropriate.

Experimental design

There is a lack of primary research studies in delusions in AD and this study fills this important gap. The design is appropriate overall though the following would be helpful:

Can the authors provide some justification for excluding cases with a CDR>3?

I would be interested to know if the authors would consider doing a sensitivity analysis that explores the severity of delusions in relation to cognitive performance. I think the NPI>=1 is reasonable but it would be interesting, in my view, to examine the raw delusion NPI score or perhaps a categorical score (0, 1-3, >4). If cognitive problems are a key substrate of delusions ideation then one might expect there to be correlations with severity of symptoms.

Could the authors provide a descriptive breakdown of number of patients with paranoid delusions and misidentifications? I don’t expect there to be enough misidentification cases to stratify the sample but this information would help with characterising the sample and drawing comparisons with past and future research.

Is anything known about the past history of delusions in the period between dementia diagnosis and the first assessment in this study?

What was the burden of the other NPI items in the delusion and no delusion groups. Did the authors consider controlling of any of these in the analysis (to rule out confounding of other symptoms)? This could be included in Table 1.
Also in table 1, please include the dementia severity of each group, the duration of dementia (time since diagnosis), and the number of patients taking antipsychotics or cognitive enhancing drugs.

Validity of the findings

A clearer statement in the abstract and in the main text that the analysis was not adjusted for multiple comparisons would be more transparent and help interpretation.

The authors mention theoretical models of delusions which for me raises questions about transdiagnostic mechanisms underlying delusions across different disorders. Could the authors comment on similarities/differences in the neuropsychological correlates of delusions in Ad vs delusions in other psychiatric disorders?

Reviewer 2 ·

Basic reporting

The present article seems clear, unambiguous and written in a technically correct form; with respect to the use of the English language, I reported some sentences that were unclear to me in the comments section.
Sufficient background is provided to demonstrate how the work fits into the present literature, but, in my opinion, more space may be given to the supposed limitations of previously published research on the issue, in order to better justify the need for a new one.
The article’s structure, figures and tables seem appropriate and always relevant to the content of the article. However, regarding figures and tables, I have provided some notes in the specific comments section.

Experimental design

The present study seems consistent with the aims and scope of the journal. The research question is clear, but, as mentioned above, I suggest to better highlight the present research's distinctiveness and contribution in comparison to the current literature.
Regarding the method section, I left a few notes in the comments section below.

Validity of the findings

In my personal opinion, the description of the results, and particularly their interpretation, could be improved. Although the reported data seem robust, statistically sound and controlled, the way they are presented and the interpretations of them are not always clear and consistent, ending up not always highlighting the strengths of the present work. More details are given in the comments section below.

Additional comments

The present study aimed to examine the impact of delusions in AD on cognitive functioning and ADL, over time, controlling for disease severity. The paper deals with a clinically important topic, given the burden that behavioural and psychological symptoms of dementia can have on the patients themselves and their caregivers. However, numerous previous studies on the topic have been already carried out and, in my opinion, the manuscript currently presents some limitations, especially in relation to the interpretations provided, that should be addressed. In particular, I have some comments for improving the quality of the article that are reported below.
1. With respect to the use of the English language, I would try to reformulate the following sentence in the introduction: “In this context, the neuropathological…and ADL” (lines 103-105). In addition to that, at the line 172 of the method section, I would specify that the term “giut” corresponds to the international phonemic alphabet phonemes of “g”.
2. In the method section, the issue surrounding the lack of informed consent is not clear to me. What does it mean “this study analysed existing test instrument”? Is it a retrospective study?
3. Moreover, I would describe the experimental procedure in more detail: when was the first evaluation performed, when was the follow-up, and for what types of patients? Regarding this, I would split the discussion of the experimental procedure from the statistical section.
4. With respect to the results section, in the introduction of the paragraph “Comparison According to the Presence of Delusion in Baseline Assessment” the absence of differences between groups at the baseline for sex, age, and education is specified, while there is no mention of any differences in AD severity between the two groups. In my opinion, this is a noteworthy data point that should be included both here and in the corresponding table.
5. In the same paragraph you say that no differences were found between the baseline and the follow-up in the non-delusion group. Have you wondered why, since it is a degenerative disease characterized by progressive cognitive impairment?
6. In the following paragraph “Comparison According to the Presence of Delusion in Follow-up Assessment” you mention that patients who manifested delusions at follow-up worsened on a semantic fluency test. It’s not clear to me whether the severity of AD was also taken into account for this comparison.
7. With respect to the discussion section, in my opinion, the shift between the worsening in the verbal immediate recall test in patients with delusions to prediction errors needs more clarity.
8. In the lines below it is assumed that patients who will develop delusions and present with greater cognitive difficulties at the first evaluation have greater brain damage. Have you considered this to be the explanation for the first finding as well? Greater delusions indicate greater neurological damage that is also reflected in worse cognitive performance. Again, I think it is important to better specify the extent to which the severity of AD was taken into account in the present study.
9. Taken together, in my opinion, the results appear to lack homogeneity, which should be taken into account. For example, why does a difference appear over time between the group without delusions and the group with delusions on the instant recall test, while the group without delusions and the group with add-delusions differ at follow up on the semantic fluency test? Or again, why are the tests that distinguish between the latter two groups, and so potentially capable of predicting the incidence of delusions, still different from those mentioned above? In my opinion these topics require further clarification and argument.
10. Furthermore, the concluding sections of the discussions seem to lack an attempt to bring together the various findings for which single and different interpretations were instead given (predictive errors, major neurological impairment, temporal lobe involvement). Also, while interesting, the insights into the possible effect of different drug treatments do not seem to be in line with what was said immediately before with respect to the association between severity of illness, delusions, and cognitive functioning. In one case, it seems that controlling for AD severity, no more differences emerge at neuropsychological tests (which emerged instead in the other studies that did not take this into account); on the other hand, it is stated that cognitive performance would somehow be protected and contained by the typical drug treatment of delusions (which, however, should also be present for the patients evaluated by the other studies). Finally, attributing such importance to SVLT-E and COWAT_animal in differentiating between patients with/without delusions seems a bit far-fetched given the lack of homogeneity of the results. Regarding the neuropsychological findings, I would suggest focusing more on the baseline differences between individuals who later acquired delirium and those who did not. These, with further research, may be helpful in predicting the occurrence of delusions.
11. Finally, with regard to the latter comments, I would consider revising the conclusions to better highlight the article's strengths, which, in my opinion, are the strong association between delusions and loss of ADL, as well as the presence of possible cognitive difficulties prior to the onset of delusions. Instead, given the low homogeneity of the data, the recruitment limitations indicated, and the discrepancies compared to the current literature, I would consider the link between delusions and cognitive functioning just as a potential area for further investigation.
12. Lastly, I leave some specific notes about the tables and figures:
• In table 1, as mentioned above, I would add the data about the severity of AD.
• Table 2 and 3 have their respective titles inverted.
• In Figure 1, it's unclear to me why the 81 patients are introduced as "follow-up visit assessment". In my opinion, it appears a little confusing as you later talk about baseline and follow-up. In addition, depending on how you change the text above, I would adjust this figure to better clarify the experimental procedure.
• In Figures 2 and 3, I would reconsider the descriptions. In both situations, it appears a bit harsh to talk about severe impairments or declines in cognitive functioning when the difference between groups was significant in only one test. In my opinion, the descriptions provided in the results section of the manuscript appeared more accurate.

---

## Round 0.2 · accepted · Accept

Dear Authors,

Thank you for your submission to PeerJ.
All the main comments and suggestions raised by the Reviewers have been addressed. Therefore, the manuscript can be accepted for publication.

Kind regards,
Marialaura Di Tella

·

Basic reporting

The authors have provided adequate responses and acknowledged limitations appropriately.

Experimental design

NA

Validity of the findings

NA

Additional comments

NA

Reviewer 2 ·

Basic reporting

no comment

Experimental design

no comment

Validity of the findings

no comment